# Development and validation of a risk prediction tool for drug-related problems in pre-operative elective surgical patients (mediPORT): A case-control study

Stephanie Clemens[1,2,3]*, Clara Simon[1], Wanda Lauth[4], Olaf Rose[1], Georg Zimmermann[4,5], Peter Gerner[3], Christina Dückelmann[1,6], Maria Flamm[2,7], Johanna Pachmayr[1,2]

**1** Institute of Pharmacy, Pharmaceutical Biology & Clinical Pharmacy, Paracelsus Medical University Salzburg, Salzburg, Austria, **2** Center of Public Health and Health Services Research, Paracelsus Medical University Salzburg, Salzburg, Austria, **3** Department of Anaesthesiology, Perioperative Medicine and Intensive Care Medicine, University Hospital of the Paracelsus Medical University Salzburg, Salzburg, Austria, **4** Team Biostatistics and Big Medical Data, IDA Lab Salzburg, Paracelsus Medical University Salzburg, Salzburg, Austria, **5** Department of Artificial Intelligence and Human Interfaces, Faculty of Digital and Analytical Sciences, Paris Lodron University Salzburg, Salzburg, Austria, **6** Landesapotheke Salzburg, Salzburg, Austria, **7** Institute of General Practice, Family Medicine and Preventive Medicine, Paracelsus Medical University, Salzburg, Austria

* stephanie.clemens@pmu.ac.at

**Data availability statement:** The authors are unable to publicly release the dataset due to ethical and legal restrictions (data contain potentially sensitive information). Researchers seeking access to confidential data can obtain it from the Paracelsus Medical University

## Abstract

### Background

Drug-related problems (DRP) in pre-operative care can harm patient outcomes. This study aimed to develop and validate a pre-operative risk prediction tool (mediPORT) to calculate the probability of DRP in admitted patients.

### Methods

Elective surgery patients aged ≥ 18 years admitted to the pre-anaesthesia clinic and participating in a medication review by pharmacists were included in this case-control study. Routinely reported patient variables were included in a backward stepwise logistic regression to determine the most relevant predictors (minimum Akaike Information Criterion) of DRP. Performances using the area under the receiver operating characteristic curve (AUC) were assessed to test the model. Internal validation was performed using a 10-fold cross-validation procedure.

### Results

The target population consisted of 11,176 participants, of whom 284 cases with ≥ 1 DRP and 980 controls without DRP were drawn. Most relevant predictors for DRP were age, number of drugs at admission, body mass index, sex and renal function.

Salzburg by meeting the required criteria. For data requests, please contact the data protection office: datenschutz@pmu.ac.at.

**Funding:** The author(s) received no specific funding for this work.

**Competing interests:** The authors have declared that no competing interests exist.

These factors were included in the final five variable model. A correlation between renal function and occurrence of DRP was found. Age and number of drugs frequently appeared in all models of the backwards elimination and represented an alternative two variable model. The AUC for predicting DRP were 0.823 (CI 95% 0.766–0.879) for the five-variable model and 0.872 (CI 95% 0.835–0.909) for the two-variable model. In the validation model, sensitivity was 77.6% and specificity was 76.5% for the five-variable model and 81.3%, 75% for the two-variable model, respectively.

## Conclusions

Resulting equations can be used by hospital admission to identify patients at high risk, for whom a precise assessment of medication is critical.

## Introduction

Patients admitted to surgery wards are at high risk for inappropriate pharmacotherapy, partially caused by transitions across the continuum of care [1]. Clinical pharmacists play a key role in identifying drug-related problems (DRP) and preventing adverse drug events (ADE). A DRP was defined by the Pharmaceutical Care Network Europe (PCNE) as "*an event or circumstance involving drug therapy that actually or potentially interferes with desired health outcomes*" [2]. Medication reviews represent an important strategy to optimize pharmacotherapy by detecting DRP and recommending potential alternatives through an interprofessional and patient-centered approach [3]. This activity is time-consuming, hence personnel shortages and limited resources impose a noteworthy barrier to clinical pharmacy interventions in medication management [4]. Moreover, the increasing pressure on healthcare services across many countries of Europe requires new and more efficient workflows. According to the World Health Organization (WHO), one approach to enhance healthcare capacity is to expand the use of digital tools [5]. Prioritizing pharmaceutical care activities can make these services more efficient and increase availability for high-risk patients. Early identification of vulnerable patients can improve medication safety, quality of care and can lead to substantial cost reductions for the healthcare system [6]. Recently published systematic reviews on prediction tools of DRP highlight the increasing interest in patient prioritization. The primary challenges with existing predictive tools include their moderate performance, limited validation and difficulty in adapting to diverse clinical settings. Additionally, there is a notable absence of straightforward, easily accessible variables that can be quickly utilized without relying on complex assessment frameworks [7,8]. Furthermore, research on risk factors, which are associated with DRP is scarce for surgical patients. Studies have evaluated outcomes of ADE and adverse drug reactions (ADR) in hospitalized patients [6,9,10]. Saldanha et al. and Lima et al. developed an internally validated risk score on the various types of DRP in adult patients admitted to a general

hospital [10,11]. According to Saldanha et al., several risk factors for DRP were identified based on patient variables at admission and drugs prescribed during the first two days of hospitalization. Key factors included an elevated heart rate and the use of multiple drugs targeting the alimentary/metabolic system, systemic anti-infectives or drugs affecting blood or its components. Patients admitted for elective surgery had a reduced risk of DRP [11]. Geeson et al. developed an internally validated prognostic model to include hospital pharmacists' input on medical wards. In this study, thirteen risk factors for DRP were identified including socioeconomic status, comorbidity and medication count, estimated glomerular filtration rate, white cell count, allergies, specific drug classes, and diagnoses. These factors were incorporated into an Excel-based tool that predicts the risk for DRP and classifies patients into three different risk categories [12]. Most studies assessed risk factors in older adults [4,13] or medical specialties such as internal medicine [9,14]. Stewart et al. and Rose et al. identified patient factors, predicting a high benefit from medication reviews in the primary care setting [15,16]. Key factors that influenced the effectiveness of medication reviews included the number of drugs, discrepancies between prescribed and actual usage, baseline Medication Appropriateness Index (MAI), and the duration of the intervention. While most of these parameters required extensive patient assessment, the number of drugs were administrable for an initial screening [16]. In a follow-up study, Rose et al. demonstrated that physicians' intuitive judgments served as a reasonably effective predictive factor. However, these judgments should be surpassed by any screening tool considered the gold standard [17]. Another study selected risk factors for DRP based on expert opinions [18]. The resulting tool, a self-administered questionnaire, was validated to classify hospitalized non-acute older patients based on their risk for DRP [19]. Developing tools to prioritize patients for pharmacotherapy optimization is complex. It faces barriers such as method selection, predictor choice, cut-off point estimations, and validation [9]. The present study endeavors to fill a knowledge gap in available tools by developing and validating a diagnostic model. This model aims to identify high-risk elective surgical patients being prone for experiencing potential and manifest DRP, which need to be reviewed more closely. The clinical value of such a DRP prediction tool is to highlight patients at risk, so that caution can be taken and DRP can be resolved, before they become significant. The model is based on routine data of medication reviews, conducted by clinical pharmacists at hospital admission. Study objectives were to: a) identify predictors for potential and manifest DRP in elective surgical patients, b) develop a multivariate score based on most relevant predictors to anticipate the risk of existing DRP and c) validate the tool in the remaining sample.

## Materials and methods

The study was conducted in line with the ethical principles of the Declaration of Helsinki and current Good Clinical Practice. The study protocol was approved by the local ethics committee of Salzburg County, Austria (ID: 1158/2021), which waived the need for written informed consent because of the retrospective nature of the study. A data analysis and statistical plan were written and posted on a publicity accessible server (German Clinical Trials Register, ID: DRKS00028763) before data were accessed.

The Transparent Reporting of a multivariable prediction model for Individual Prognosis Or Diagnosis (TRIPOD) checklist was used for reporting the development and validation of the pre-operative risk prediction tool for medication review (mediPORT) (S1 Table) [20]. Study objectives were created by using the population, intervention/exposure, control intervention/exposure and outcome (PICO) framework:

a) Population: adult patients, admitted to a pre-anaesthesia clinic (PAC) with an elective surgery;

b) Intervention: exploration of predictor variables in patients with DRP (cases);

c) Comparison: exploration of predictor variables in patients without DRP (controls) and

d) Outcome: risk for DRP (or not).

## Study design

The study design was a case-control study (Fig 1). Patient data consisted of clinical routine data, collected from electronic health record charts of a tertiary care university hospital in Salzburg, Austria between January 1 and December 31, 2021. Cases were defined as elective surgical patients with one or more DRP identified by pharmacists through a comprehensive medication review (PCNE type 3 review) and documented in the hospital information system [3]. Controls, all of whom were reviewed by clinical pharmacists, were defined as elective surgical patients without any identified and documented DRP. Initially, the ratio of controls to cases was set to 4:1 as statistical power strongly increases within this ratio [21] and the prevalence of DRP in the target population was low [22]. A prior study investigating secondary outcomes including the nature and prevalence of DRP in the target population revealed that 9.89% patients experienced at least one DRP. The primary causes of DRP were identified as drug-drug interactions (30.3%) and supra-therapeutic dosing (18.0%) [22]. Given this relatively low prevalence, it was anticipated in the study planning that the number of eligible cases would be limited. Therefore, a 1:4 case-control ratio was selected not only for statistical efficiency but also to ensure a sufficient total sample size to power the analysis. This ratio allowed for better estimation of exposure-outcome associations and model parameters without the need to include all available cases.

## Setting

The pre-anaesthesia clinic (PAC) represents a specialty unit, located within the university hospital Salzburg, where patients are assessed before surgery. Evaluation includes interviews, examinations, a review of the medical, surgical and anaesthesia anamnesis, a review of current medication and preoperative tests. According to clinical routine at PAC, patients were reviewed by one out of nine clinical pharmacists, who were trained in medication reconciliation and comprehensive medication review according to the PCNE type 3 definition [3] and by anaesthetists who performed clinical assessments regarding the delivery of anaesthesia care for surgery procedures. Anaesthetists gave the final approval to the operation. Fig 2 describes the collaborative process at admission in more detail.

## Participants and sample size

Inclusion criteria were age ≥ 18 years, both sexes (defined as sex assigned at birth), admission in 2021 at the PAC for elective surgery, participation in a medication review and hospitalization for more than 24 hours to one of the nine different surgical specialty clinics:

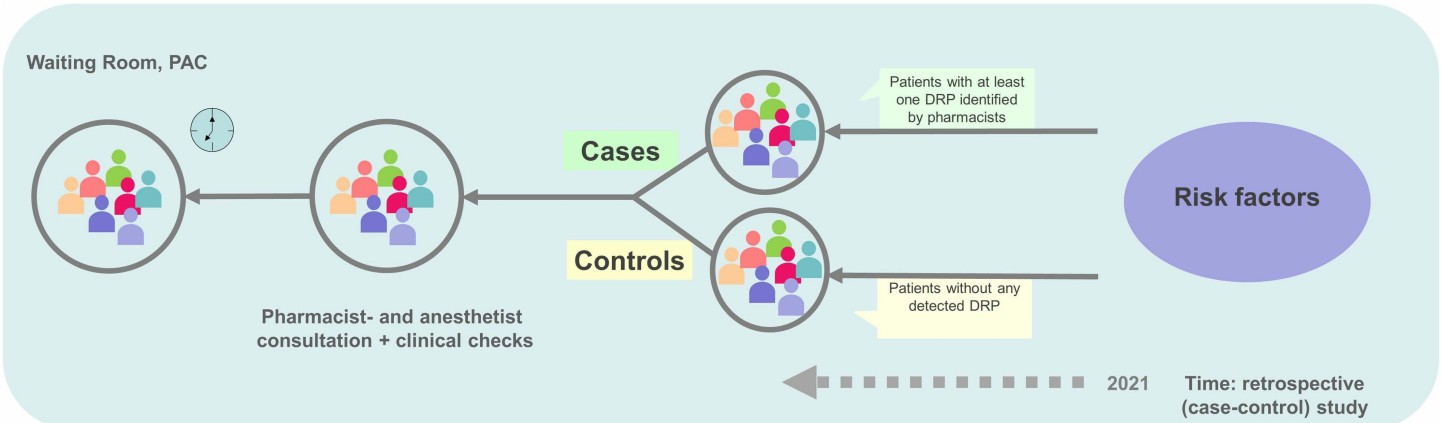

**Fig 1. Design of the case-control study.** PAC: pre-anaesthesia clinic, DRP: drug-related problem.

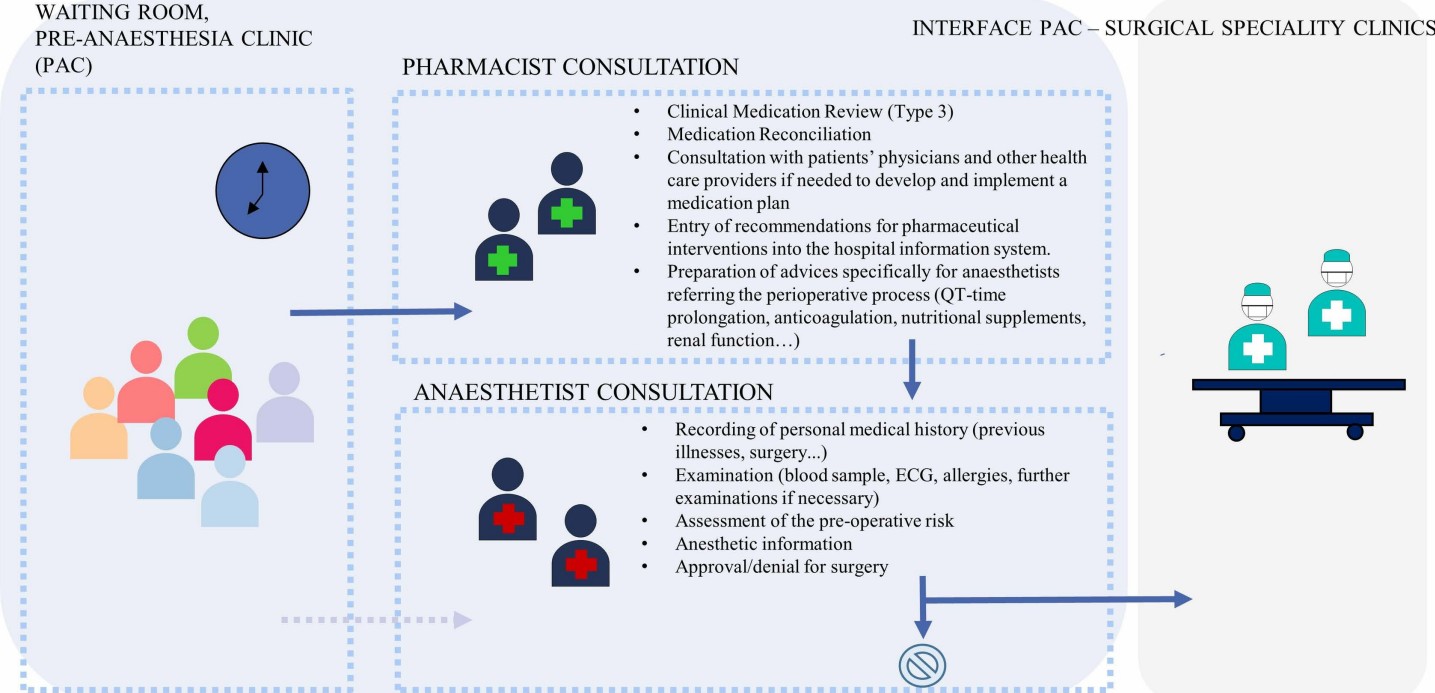

**Fig 2. Pre-anaesthesia clinic (PAC), routine care procedure.** ECG: Electrocardiography.

- cardiac surgery, vascular surgery and endovascular surgery (GF)

- dermatology/allergology (DE);

- ear, nose and throat diseases (LA);

- general surgery (CH);

- gynecology and obstetrics (GY);

- oral and maxillofacial surgery (KG);

- orthopedics and traumatology (OT);

- special gynecology (SF) and

- urology and andrology (UR).

Patients receiving day care surgery at the remaining wards were excluded from the study as they did not undergo consultation by pharmacists.

According to a population-based case-control study design, cases and controls were sampled from a single source population (n = 11,176) between January 1, 2021 (01.01.2021), and December 31, 2021. To reach the target case-control ratio of 1:4, a random selection of 1200 patient controls was carried out. Consequently, the target sample size (i.e., including cases and controls) was set to 1500 patients. This sample size allowed to estimate the sensitivity of the multivariable model to a precision of plus/minus 5% in terms of the width of a 95% CI, under the assumption that the sensitivity was at least 75%. Moreover, the number of cases was also adequate with respect to the generally recommended minimum event-per-variable (EPV) ratio of 10 [20].

## Endpoint and predictors

The primary outcome was the exposure to a potential or manifest DRP at patients' admission to PAC, detected through a medication review. Patients' charts and pharmacist records were reviewed to determine the clinical outcome. There was no blind assessment of the predicted outcome.

Predictors selection was conducted based on a) a focused literature review using PubMed and Google Scholar b) clinical knowledge and experience in case of identifying objective and easily ascertainable factors, which were applicable on clinical practice data and c) consultations with experts from different fields, including anesthesiology, family medicine, geriatrics, pharmacotherapy and clinical pharmacy. The following factors were applied when determining the final set of variables (predictors): reliability, consistency, availability, and costs of predictor measurement relevant to the targeted set. Steps a-c were interdependent and were performed complementary in several loops.

Twelve potential candidate variables were considered for inclusion in the mediPORT development and are described in detail in the S1 Appendix:

1) age;

2) sex;

3) main residence;

4) body mass index (BMI);

5) intolerance;

6) allergy;

7) pre-operative risk, according to the American Society of Anesthesiologists (ASA) [23];

8) Charlson Co-morbidity Index (CCI) [24];

9) hospital specialty clinic for elective surgery;

10) hospitalization within the previous 12 months;

11) renal function [25] and

12) number of drugs at admission.

Predictors were obtained from the hospital's electronic information system ORBIS (Dedalus HealthCare, Bonn, Germany) and included patients' demographic characteristics, medical history, physical examination, disease characteristics, routine laboratory test results and previous treatments. A licensed pharmacist (SC) and a master pharmacy student (CS) independently extracted and assessed patient data retrospectively between May 1, 2022 and May 31, 2023 for research purposes. In regular meetings, extracted and analyzed data were compared and checked to its completeness to validate the process. Occurred inconsistencies and cases of doubt were discussed and resolved.

## Missing data

Numbers of participants for cases and controls with missing values were assessed. There were no missing data for sex, age and hospital ward. Information on BMI, intolerance, allergy, number of drugs at admission, ASA and residence were missing in ≤ 1% of the participants each. These rates were considered low. Complex classification of CCI and search in patient history (hospitalization during the last 12 months) were stopped after assessment of 70% of participants because of time issues. Consequently, missing values of the remaining two variables resulted in 30%. Incomplete data of renal

function was 16.3%. Main reason was skipping the laboratory test in routine care in accordance with the ASA Task Force on Paranesthesia Evaluation recommendations These guidelines suggest limited testing to reasonable cases where it is likely to yield clinical benefits, such as improving the safety and effectiveness of anesthetic processes, particular for young and healthy patients [26]. Variables were not imputed, as it was not possible to justify a missing at random assumption. However, if the data for the key predictors identified in the model development procedure (see below) was complete, these patients were included in the test data set.

## Statistical analysis

To determine the predictors and describe the characteristics of the sample, the patients' characteristics were first presented descriptively by calculating the minimum, first quartile, median, third quartile, and maximum for ordinal variables and, in addition, the mean and standard deviation for metric variables. Nominal variables were summarized using absolute and relative frequencies. To identify collinearities, the correlation coefficient between the metric and/or ordinal variables was considered. In addition, two-sample T tests using the Welch approximation for unequal variances for metric, Fishers exact tests for nominal, and the Mann-Whitney U test for ordinal variables were used to statistically examine differences between those patients with and without DRP. These results also served as the basis for identifying consistencies and inconsistencies in the subsequent construction of the regression model to predict the occurrence of DRP, based on the independent variables. Because of the large number of related tests, the Bonferroni-Holm method was used to correct for multiplicity.

## Model development

Subsequently, for the logistic regression models predicting DRP, a backward variable selection algorithm was used, with the Akaike Information Criterion (AIC) value being used to determine the best model [27]. Because this procedure requires complete data, patients with missing data were excluded from the model development. However, if there were no missing values in the variables that were included in the final model, the data was still used for the test dataset to validate the final model. The remaining data set (without missing values) was divided by randomization into 70% training and 30% test data. Depending on the composition of the training and test set, the variables that were finally selected for inclusion in the prediction model may vary. Therefore, in order to achieve a good generalization and compensate for possible instabilities (e.g., overfitting, underfitting), the split of the data set and the subsequent selection of variables by backward elimination was repeated several times (1000x) [27]. At the same time, the EPV ratio was computed for each run to ensure that the resulting models at least respected the rule of thumb EPV > 10, when estimating the parameters in the case of different compositions of the training set to avoid additional overfitting [20]. Those predictors that appeared in more than 50% of the backward eliminations were finally used to build the resulting model. These variables are also the ones that occur as combination in most of the models with the lowest AIC per run. In addition, a second model was created with those variables that occurred in 100% of the cases.

## Model validation

Receiver operating characteristic (ROC) analysis was performed to validate the models and select the best cut-off value. In addition, accuracy, sensitivity, specificity, and area under the curve (AUC) were determined using the test set on the one hand and 10-fold cross-validation on the other [27,28]. As part of the cross-validation, a calibration plot was also created to evaluate the predicted probability versus the frequency of occurrence, using different compositions of test and training data sets.

Boxplots, barplots as well as the ROC curves were created for visualization. The two-sided significance level $\alpha = 0.05$ was used. All analyses were carried out using the statistical software package R [29].

## Results

### Participants

During the year 2021, there were 11,176 admissions for elective surgery. Of these, 1200 patients without a documented DRP were selected at random and 300 patients with a DRP, meeting the inclusion criteria, were included. Patients, who did not undergo review by clinical pharmacists, i.e., undergoing day care surgery and under the age of 18 years, were excluded from the study. The flow of the participants through the study is shown in Fig 3. Due to the large sample size, risks of selecting spurious predictors (overfitting) and failing to include important predictors (underfitting) were low [20]. EPV of the full sample (300 cases/12 variables) resulted in an anticipated EPV of 25, exceeding the "rule of thumb" of ≥10 EPV. Therefore, even though the actual number of cases included in the training set will be somewhat lower, due to the split into training (70%) and test (30%) data, the EPV will be well above 10.

Most individuals were from the urban region (56.8%) and of male sex (53.4%). BMI was higher for cases as compared to controls, 27.68 kg/m² (SD 5.92) versus 26.47 kg/m² (SD 5.72). In aspects of polypharmacy, number of drugs at admission was significantly higher in cases remaining a mean of 8.58 drugs per patient (SD 4.22) than in control patients with a mean intake of 2.77 drugs per patient (SD 3.48). A summary of sociodemographic and clinical data is listed in Table 1. Boxplots and barplots of participant characteristics are shown in the S2 Appendix.

### Development of the mediPORT risk model

Out of the twelve selected predictors, ten variables showed a significant difference in prevalence between DRP and non-DRP patients: sex (p<0.001), age (p<0.001), BMI (p=0.007), intolerance (p<0.001), number of drugs at admission (p<0.001), CCI (p<0.001), ASA (p<0.001), hospitalization in the last 12 months (p<0.001), hospital ward (p<0.001) and renal function (p<0.001). No multicollinearity was demonstrated in the model, which enabled reliable statistical inferences. The final EPV value related to the training set was mean (SD) 15.59 (±0.49) and thus fulfilled defined EPV criteria.

The backwards elimination using the AIC resulted in a final risk prediction model of five variables (5VM): age, number of drugs at admission, BMI, sex and renal function. Correlations and a web code for the variable selection are shown in the S2 Table. Noticeably, age and number of drugs occurred in all models (100%) of the backwards elimination, and were used to calculate a simpler model of two variables (2VM). The identified variables were subsequently used to form the final mediPORT risk model (S3 Table).

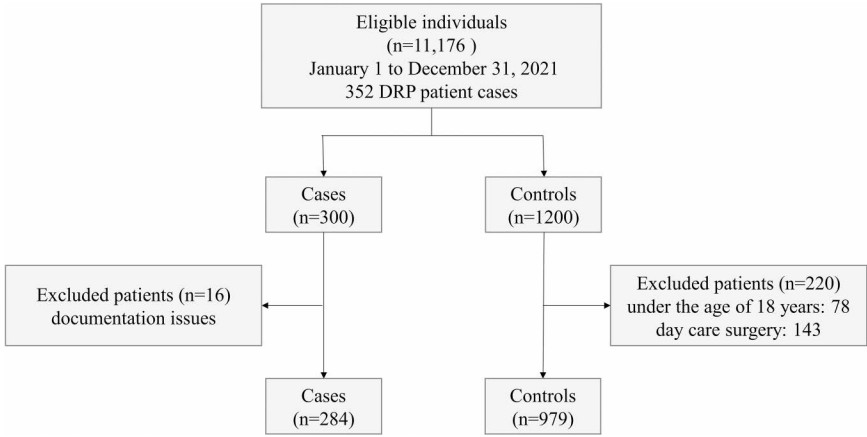

**Fig 3. Participant flow diagram.** DRP: Drug-related problem.

**Table 1. Participant characteristics. Frequencies, proportions, means, SDs and OR are presented. Not applicable (NA): Missing information due to lack of data in clinical records. BMI: body mass index, DRP: drug related problem, CCI: Charlson Comorbidity Index, ASA: American Society of Anesthesiologists, OR: odds ratio, SD: standard deviation.**

| Variables | Categories | Patients with DRP (cases) | Patients without DRP (controls) | Total sample | OR (CI 95%) | P value |
|---|---|---|---|---|---|---|
| Sex (n, %) | Female | 105 (36.97) | 484 (49.44) | 589 (46.66) | 1.666 (1.26 to 2.21) | 0.0016 |
| | Male | 179 (63.03) | 495 (50.56) | 674 (53.37) | | |
| | NA | 0 (0) | 0 (0) | 0 (0) | | |
| Age (years; mean, SD) | | 72.41 (10.78) | 54.67 (19.03) | 58.66 (19.02) | | < 0.001 |
| | NA (n, %) | 0 (0) | 0 (0) | 0 (0) | | |
| BMI (mean, SD) | | 27.68 (5.92) | 26.45 (5.78) | 26.72 (5.83) | | 0.008 |
| | NA | 4 (1.4) | 8 (0.82) | 12 (0.01) | | |
| Intolerance (n, %) | Drugs | 76 (26.76) | 171 (17.47) | 247 (19.56) | | 0.0035 |
| | Other | 33 (11.62) | 177 (18.08) | 210 (16.63) | | |
| | No | 174 (61.27) | 620 (63.33) | 794 (62.87) | | |
| | NA | 1 (0.35) | 11 (1.12) | 12 (0.95) | | |
| Allergy (n, %) | Yes | 109 (38.38) | 348 (35.55) | 457 (36.18) | 0.896 (0.676 to 1.19) | 1.0000 |
| | No | 174 (61.27) | 620 (63.33) | 794 (62.87) | | |
| | NA | 1 (0.35) | 11 (1.12) | 12 (0.95) | | |
| Number of drugs at admission (mean, SD) | | 8.58 (4.22) | 2·77 (3.48) | 4.09 (4.39) | | < 0.001 |
| | NA (n, %) | 0 (0) | 5 (0.51) | 5 (0.40) | | |
| CCI (mean, SD) | | 2.6 (2.18) | 1·34 (1.89) | 1.74 (2.07) | | < 0.001 |
| | NA (n, %) | 0 (0) | 383 (39.1) | 383 (30.32) | | |
| ASA (n, %) | 1 | 1 (0.35) | 250 (25.55) | 251 (19.87) | | < 0.001 |
| | 2 | 71 (25.00) | 422 (43.11) | 493 (39.03) | | |
| | 3 | 199 (70.10) | 291 (29.72) | 490 (38.80) | | |
| | 4 | 13 (4.56) | 15 (1.53) | 28 (2.22) | | |
| | NA | 0 (0) | 1 (0.10) | 1 (0) | | |
| Hospitalization during last 12 months (n, %) | Yes | 216 (76.1) | 321 (32.79) | 537 (42.52) | 0.368 (0.264 to 0.509) | < 0.001 |
| | No | 68 (23.9) | 275 (28.09) | 343 (27.16) | | |
| | NA | 0 (0) | 383 (39.12) | 383 (30.32) | | |
| Hospital ward (n, %) | CH | 26 (9.15) | 145 (14.81) | 171 (13.54) | | 0.0035 |
| | DE | 13 (4.58) | 42 (4.2) | 55 (4.36) | | |
| | GF | 57 (20.07) | 63 (6.44) | 120 (9.50) | | |
| | GY | 12 (4.23) | 118 (12.05) | 130 (10.29) | | |
| | KG | 9 (3.17) | 89 (9.09) | 98 (7.76) | | |
| | LA | 29 (10.21) | 146 (14.91) | 175 (13.86) | | |
| | OT | 58 (20.42) | 147 (15.02) | 205 (16.23) | | |
| | SF | 12 (4.23) | 52 (5.31) | 64 (5.07) | | |
| | UR | 68 (23.94) | 177 (18.01) | 245 (19.40) | | |
| | NA | 0 (0) | 0 (0) | 0 (0) | | |
| Residence (n, %) | R | 125 (44.01) | 415 (42.39) | 540 (42.76) | 1·056 (0.802 to 1.392) | 1.0000 |
| | U | 159 (56.00) | 558 (57.00) | 717 (56.78) | | |
| | NA | 0 (0) | 6 (0.61) | 6 (0.47) | | |
| Renal function (n, %) | <15 (3) | 9 (3.17) | 14 (1.43) | 23 (1.82) | | < 0.001 |
| | 15-29 | 14 (4.93) | 14 (1.43) | 28 (2.22) | | |
| | 30-59 | 105 (36.97) | 82 (8.38) | 187 (14.81) | | |
| | 60-89 | 97 (34.15) | 360 (63.77) | 457 (36.18) | | |
| | >89 | 47 (16.55) | 315 (32.18) | 362 (28.66) | | |
| | NA | 12 (4.22) | 194 (19.82) | 206 (16.31) | | |

*(Continued)*

**Table 1.** (Continued)

| Variables | Categories | Patients with DRP (cases) | Patients without DRP (controls) | Total sample | OR (CI 95%) | P value |
|---|---|---|---|---|---|---|
| Number of DRP (mean, SD) | | 1.43 (0.75) | 0 (0) | 0.32 (0.69) | | NA |
| | NA (n, %) | 0 (0) | 0 (0) | 0 (0) | | |
| Duration of hospital stay (mean, SD) | | 8.87 (8.85) | 6.83 (7.54) | 7.3 (7.9) | | < 0.001 |
| | NA (n, %) | 14 (4.93) | 72 (7·35) | 86 (6.81) | | |
| Approval of surgery (n, %) | Yes | 218 (76.76) | 882 (90.10) | 1100 (87.10) | | < 0.001 |
| | No | 64 (22.54) | 93 (9.50) | 157 (12.43) | | |
| | NA | 2 (0.70) | 4 (0.41) | 6 (0.48) | | |

To simplify the presentation of the final mutlivariate regressions, the calculation of the individual risk was conducted in two steps. First, the risk score was calculated (either with five or two variables, depending on available variables). Next, this risk score was inserted into the exponential formula below, which finally gave the estimated risk as a percentage. This formula resulted from the conversion of the log odds of the logistic regression.

$$5VM \text{ equation for risk score}: -7{,}29712977153007 - 0{,}394421234628523 \times female(yes; no)^*$$
$$+ 0{,}423965922272388 \times age + 0{,}257237146449912 \times number\ of\ drugs$$
$$+ 0{,}0339923294039108 \times BMI + 0{,}86186079010242$$
$$\times renal\ function 15-29(yes/no) + 1{,}78880673311741$$
$$\times renal\ function 30-59(yes/no) + 0{,}862978252896248$$
$$\times renal\ function 60-89(yes/no) + 1{,}26877761969504 \times renal\ function > 89(yes/no)$$

$$2VM \text{ equation for risk score}: -5{,}8184055 + 0{,}423965922272388 \times age + 0{,}257237146449912 \times number\ of\ drugs$$

$$Individual\ estimated\ risk\ of\ DRP\ (\%): \frac{e^{risk\ score}}{1 + e^{risk\ score}} \times 100$$

*1 if "yes", 0 if "no"

A modified real-life example and the formula of the models (5VM, 2VM) to illustrate the use of the prediction tool are shown in Fig 4.

## Renal function

Levels of renal function showed different risks for DRP related to CKD stage 1–5 (according to Clinical Practice Guideline for the Evaluation and Management of Chronic Kidney Disease guideline 2012) in the 5VM [25]. The highest risk of DRP was related to patients with an eGFR between 30 and 59 ml/min/1.73m$^2$ (CKD stage 3). Risk development for DRP was further related to sex and showed reduced risks for females. A curve to show the change through different levels of renal function and sex in both models on basis of the before mentioned patient case example is shown in Fig 5.

## Performance of the mediPORT risk model

Accuracy and ROC with AUC of the 5VM were evaluated by calculating the sensitivity (true positive) and specificity (true negative) of the model. The 5VM showed a sensitivity of 78.8%, a specificity of 73.6%, and an overall accuracy of 74.0%, resulting in an AUC of 0.823 (95% CI 0.766–0.879).

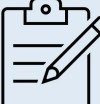

**CASE EXAMPLE**

Mrs. M is 82 years old and was admitted to the PAC for elective surgery due to her spine fracture. She lives together with her husband. Mrs. M has a medical history of arterial hypertension, arterial flutter, chronic gastritis, degenerative arthrosis, venous insufficiency and mitral valve insufficiency grade II. Her medications at admission include apixaban, hydromorphone, pantoprazole, amiodarone, lercanidipine, pregabaline, bisoprolol and valsartan. Her BMI is 23 kg/m². The last renal function was at an eGFR of 50 ml/min/1.73m².a

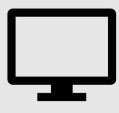

**FORMULA**

**5VM equation for risk score:** -7,29712977153007 - 0,394421234628523 x *female(yes /no)\** + 0,0423965922272388 x *age* + 0,257237146449912 x *number of drugs*+ 0,0339923294039108 x *BMI* + 0,86 1860790 10242 x *renal function 15-29(yes/no)* + 1,78880673311741 x *renal function 30-59(yes/no)* + 0,862978252896248 x *renal function 60-89(yes/no)* + 1,26877761969504 x *renal function >89(yes/no)*

**2VM equation for risk score**: -5,81840546825991 + 0,0476772991258964 x *age* + 0,267574180027192 x *number of drugs*

**Individual estimated risk of DRP (%):** $e^{risk\ score} /(1+ e^{risk\ score})$

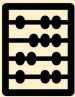

**RISK CALCULATION**

**5VM Risk score:** -7,29712977153007 - 0,394421234628523 x *1* + 0,0423965922272388 x *82* + 0,257237146449912 x *8* + 0,0339923294039108 X *23* + 0,86186079010242 X *0* + 1,78880673311741 X *J* + 0,862978252896248 X *0* + 1,26877761969504 X *0*
Individual estimated risk of DRP (%): $e^{risk\ score} /(1+ e^{risk\ score})$
**Mrs. M's absolute risk of experiencing a DRP requiring pharmaceutical care in time of hospital admission is estimated at <u>60.19%</u>.**

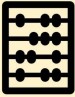

**2VM Risk score:** -5,81840546825991 + 0,0476772991258964 x *82* + 0,267574180027192 x *8*
Individual estimated risk of DRP (%): $e^{risk\ score} /(1+ e^{risk\ score})$
**Mrs. M's absolute risk of experiencing a DRP requiring pharmaceutical care in time of hospital admission is estimated at <u>55.77%</u>.**

**\****yes = 1; no = 0*

**Fig 4. MediPORT tool.** Calculation of patient risk for developing a drug-related problem requiring pharmaceutical care at hospital admission. 5VM: 5 Variable Model, 2VM: 2 Variable Model, DRP: drug-related problem, PAC: pre-anesthesia clinic, BMI: body mass index, eGFR: estimated glomerular filtration rate.

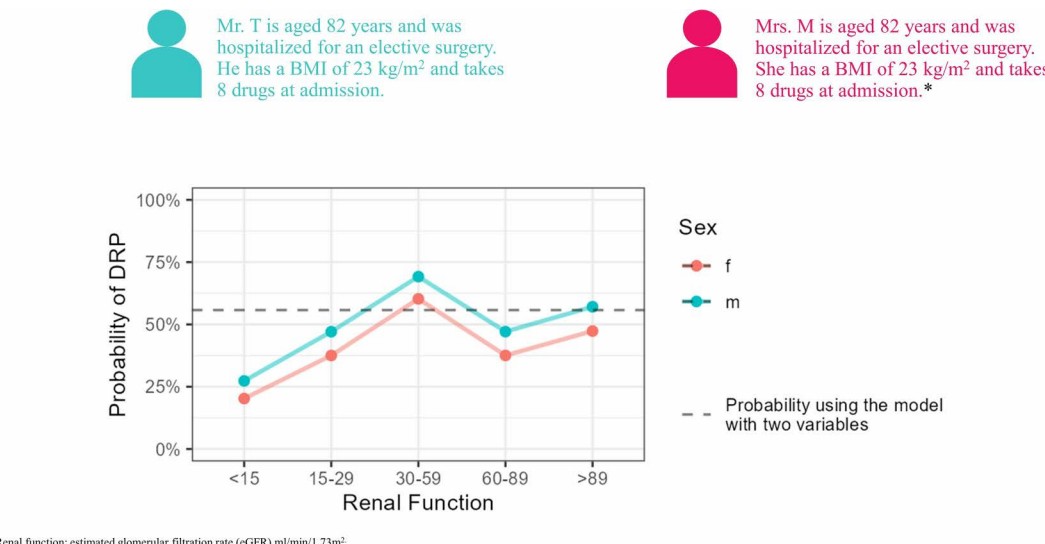

Renal function: estimated glomerular filtration rate (eGFR) ml/min/1.73m².

*In the equation everything stays the same but gets multiplied by exp(female).

**Fig 5. Renal function.** Risks of developing a drug-related problem among different renal function levels and sex within the 2VM and 5VM. f: female, m: male, DRP: drug-related problem.

The 2VM model was chosen because the backwards elimination process consistently included the variables age and number of drugs in every iteration. The 2VM was found to have an AUC of 0.872 (95% CI 0.835–0.909) with 83.7% sensitivity, 79.5% specificity and 79.2% accuracy. The ROC curves of both models are displayed in Fig 6.

## Validation

For validation, a 10-fold cross-validation was conducted. Again, only the dataset with no missing values was used. The 10-fold cross-validation used as the test set to calculate specificity, sensitivity and the ROC curve.

The 5VM demonstrated a sensitivity of 77.6% (SD 0.078) and specificity of 76.5% (SD 0.061), along with an AUC of 0.856 (SD 0.040). These values were obtained by setting the cut-off for risk of developing DRP to a model-predicted probability value of 0.35. The cut-off value was selected from a range from zero to one on the basis that there was a slight preference to obtain a high sensitivity to predict occurrence of DRP.

For the 2VM, 10 times cross-fold validation yielded in 81.3% (SD 0.071) sensitivity and 75.0% (SD 0.063) specificity with a determined cut-off value of 0.35. AUC resulted in 0.847 (SD 0.043). The ROC curves of both models including cut-off values are displayed in Fig 7. The range of cut-off values for both models is provided in the S4 Table. All results for the mediPORT tool are based on the chosen cut-off value of 0.35 for both models.

Performances of both models (2VM, 5VM) did not decrease substantially between the development and internal validation.

## Comparison of the performance

As more than one model was developed on the same data sets within the study, performances of the 5VM and the 2VM were compared. A calibration plot was used to compare predicted probability with the observed proportions of DRP (Fig 8).

## Discussion

### Summary

Within the mediPORT study, an algorithm was created that can predict probability of potential and manifest DRP in elective surgical patients at the time of hospital admission, based on routine parameters. The study supports the ASA Task Force recommendation that the assessment of risks regarding patients' medical conditions and therapies are contributing

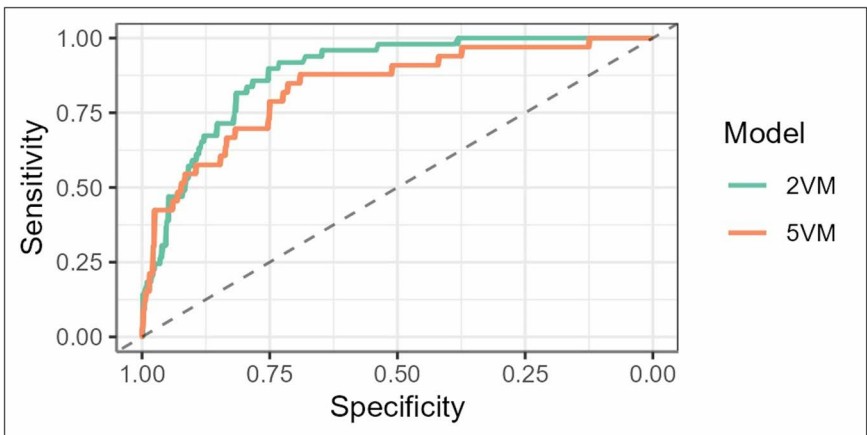

**Fig 6. Receiver operating characteristic (ROC) curves of the predictive models obtained in the training sets.** 2VM: 2 Variable Model, 5VM: 5 Variable Model.

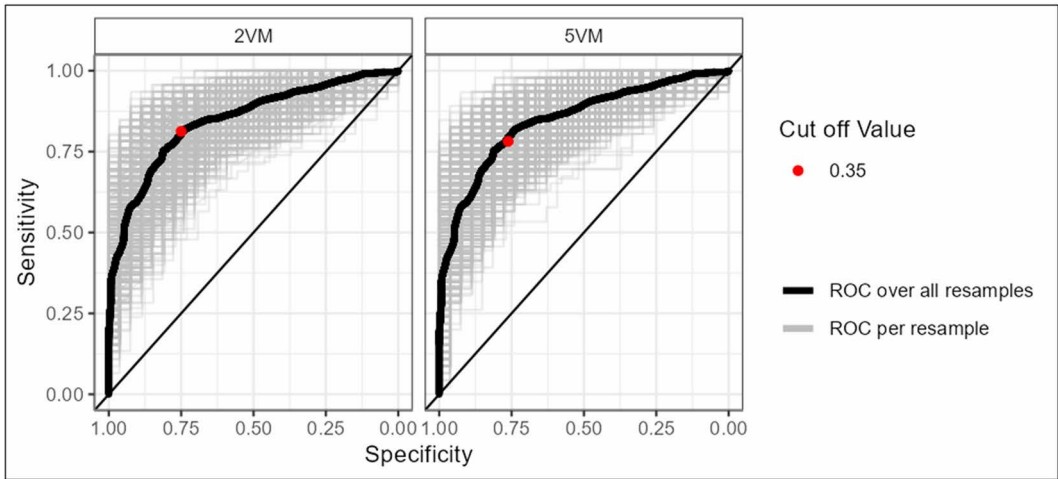

**Fig 7. Receiver operating characteristic (ROC) curves of the predictive models obtained in the validation sets.** 2VM: 2 Variable Model, 5VM: 5 Variable Model.

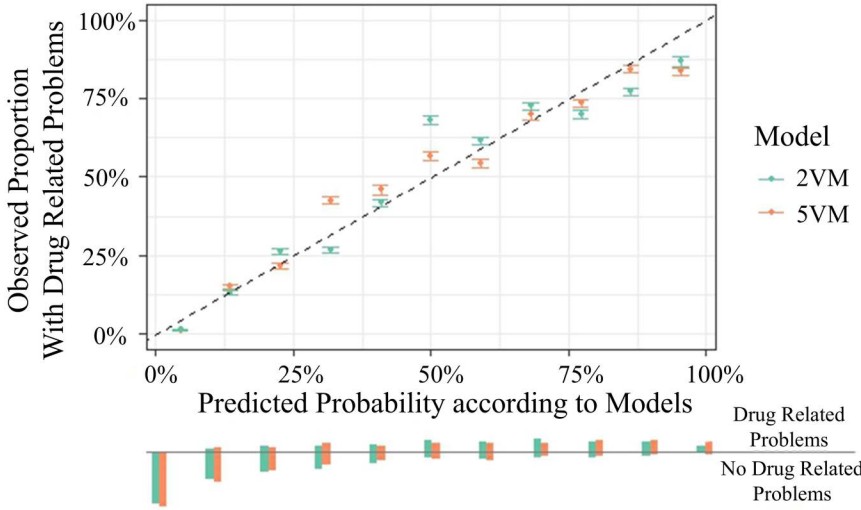

**Fig 8. Calibration plot of the mediPORT tool.** Distribution of predicted probabilities for patients with and without the outcome (drug-related problems). 2VM: 2 Variable Model, 5VM: 5 Variable Model.

to safety of perioperative care, ideal resource use and improved patient outcomes. The model was developed with an inter-professional team based on real life data and on international standards. Identified parameters consisted of demographic (age, sex), health-related (BMI), medication-specific (number of drugs at admission) and biochemical (renal function) factors, reflecting the complexity of developing DRP and risk stratification.

## Clinical implications and future research

In diagnostic modelling for identifying DRP in elective surgical patients, certain predictors emerged as highly influential. Notably, age and number of drugs at admission served as robust predictors in the 2VM and the 5VM model. The alternative

2VM model, a simplified version of the 5VM, offers practicality in situations where information required for the comprehensive model (5VM) is unavailable or time constraints are critical. Predictors of the 2VM finds resonance in previous research, notably in a study, which identified medication errors in a surgical setting [30]. However, it is essential to note that the streamlined nature of the 2VM may introduce risks of underfitting, due to its limited variety of predictors potentially compromising its reliability. Thus, the choice between the 2VM and the more comprehensive 5VM is pivot, with considerations extending beyond mere convenience. While the 2VM may suffice in certain scenarios, particularly where data availability is constrained, the 5VM (additionally including BMI, gender and renal function) offers greater depth and robustness, enhancing the model's predictive power. Hence, prudent decision-making necessitates a nuanced understanding of the trade-offs between simplicity and predictive accuracy, advocating for the utilization of the 5VM whenever feasible.

The good performances of the mediPORT versions (5VM, 2VM) with an AUC = 0.856 and an AUC = 0.847 respectively might help in medical and pharmaceutical decision making and assist the health care service at surgical admissions (i.e., effective planning and quality management). Both strengths might lead to improved medication and safety of care for surgical patients. Performances of mediPORT versions are on par with the performance of other DRP models. Høj et. al. and Saldanha et al. developed a DRP prediction tool for hospitalized patients which showed a slightly lower discriminatory power of an AUC = 0.70 and AUC = 0.65 respectively in the validation set [11,31].

The effect on type I and type II errors depends on the diagnostic models and the threshold for classification (cut-off). Considering sensitivity, specificity, and threshold are crucial when evaluating predictive model performance and real-world implications. Choosing an optimal cut-off value is challenging as a model with higher sensitivity will report lower specificity and vice versa. While patients identified as false positive (type I error) may result in increased medication scrutiny, the information on risk has potential to cause unnecessary concern to patients and clinicians. Additionally, unnecessary testing might cause a delay of the elective surgery. Nevertheless, for a DRP risk score, sensitivity (reduction of false negatives, type II error) is more important as misclassification of a high-risk patient may lead to harm [32]. Consequently, selecting a cut-off value of 0.35 for each model resulted in achieving sensitivities exceeding 77% and specificities surpassing 73% in the development and validation sets of both models. Previous developed models showed reasonable specificity but lacked in sensitivity, which was lower than 62% [10,31]. The cut-off of 0.35 for the ROC curve represents the threshold value used to classify patients into different categories based on the predicted risk of DRP. In practical terms, it means that if a patient's predicted risk score is above 0.35, they would be classified as high risk for DRP, while those with a score below 0.35 would be classified as low risk.

In modern healthcare, it is essential to base the practical use of tools on the best available evidence. This means applying evidence on test performance with careful judgment and testing efficiently to optimize patient care. Moreover, considering the value and affordability of a test before requesting it, ensures resources are used effectively. It is also crucial to be aware of the downsides and drivers of over-diagnosis, confront uncertainties, and maintain a patient-centred approach. Addressing ethical issues and recognizing cognitive biases further enhances the quality of testing. Teaching and learning these core principles through a systematic approach ensures that healthcare professionals can navigate the complexities of testing with confidence and precision [33]. When using the mediPORT tool, clinical pharmacists can intervene during medication reconciliation and patient interviews at the PAC visit to rectify any omissions.

While predicting potential DRP provides valuable insights, it is essential to consider the clinical context and the likelihood of the predicted event to actually occur. For example, even if a patient is predicted to experience a potential DRP, such as requiring a dosage adjustment for a drug due to renal insufficiency, it may not be clinically relevant in this specific case. While diagnostic models can inform clinical decision-making, their application must be considered in conjunction with the clinical context and patient-specific factors.

Future research might evaluate the predictive performance of the algorithm in a different setting. For instance, risks of patients can be assessed retrospectively by the tool and compared to the estimated risks by pharmacists in real life practice. Results can then be used to verify sensitivity (true positive) and specificity (false positive) of the tool.

## Outcomes in the context of previous research

Most of the variables tested in the tool have been previously shown to be associated with the development of DRP or have been beneficial in medication management [34,35]. Only few studies have revealed BMI as a risk factor so far [34,36]. Surprisingly, the risk for developing a DRP by the mediPORT tool varied between different CKD stages with the highest risk for CKD stage 3 and lowest risk for stage 5. Previous studies showed the risk for DRP continuously increasing among the severity of renal function. In contrast to the mediPORT study results, Garin et al, identified patients with an eGFR of <30ml/min/1.73m2 (CKD stage 4 and 5) being at highest risk for DRP [34]. O'Connor and colleagues identified patients starting being at risk with an eGFR ≤ 60 ml/min/1.73m2 for ADR (CKD stage 3 and above) [37]. Rose et al. and Falconer et al. did not find any risks associated with renal function [6,16]. Results on different risks in renal function of the mediPORT study could be interpreted in a way that patients at CKD stage 5 might be considered a special group since they require dialysis and thus are monitored closely, which might decrease the risk for occurrence of DRP. A recent systematic review on DRP in hospitalized patients with CKD revealed that starting at CKD stage 3, dose adjustment is needed in 0.4–1.7 drugs per patient [38]. This is in line with the cut-off value of O`Connor et al. who also determined the eGFR starting at 60 ml/min/1.73m2 and below as being the critical point of developing DRP [37]. In light of existing data, this study adds the information that patients with a normal and with a severely impaired renal function are at lower risk for DRP than those with a moderately reduced renal function, which frequently requires dose adjustments. This in-depth analysis can explain why previous studies have retrieved controversial results, and therefore did not identify kidney function as a risk factor. However, individual risks might always exist in patients taking certain drugs, i.e., antibiotics, oral antidiabetics or anticancer drugs, which have not been investigated in the present study [38,39]. Future analyses stratified by drug class or pharmacologic group could provide greater insight into which medications drive DRP risk in moderate versus severe CKD. Understanding these nuances may ultimately improve risk stratification and prevention strategies in clinical practice.

## Strengths and limitations

Major strengths of this study were a) the large sample size, b) the strong variety of included patients according to age (18–101 years) and hospital sites (nine different wards), c) the use of routinely collected patient data and d) the use of standardized classification systems (e.g., CKD stages, ASA classification, CCI, BMI).

Due to the low prevalence of DRP in the total elective surgery population, the case-control design of this study represents a beneficial, time- and resource- efficient approach in model development, compared to the more frequent use of prospective cohort designs. Additionally, due to the retrospective nature of the study, determined predictors could not influence the outcome assessments and thus did not cause any bias.

Nevertheless, the study has several limitations. Data were obtained from a single hospital, which limits generalizability of the study results. The data related entirely on archived patient records. Some data were not archived/available in digital or written form and thus were not included in the dataset. This might have led to underreporting. The identification of DRP depended on the different experience of the clinical pharmacists' skills, but were conducted by dual control principle.

## Conclusion

Age, sex, number of drugs, BMI and renal function represent five independent predictors, which can be used to aid manifest and potential DRP identification in elective surgical patients. Additionally, the mediPORT tool offers an alternative approach with two independent predictors (age and number of drugs) in case of limited patient information. The tool provides two robust and internally validated models, which strongly follow the idea to enhance medication safety and quality of care by digitalization. Pharmacists can be utilized in a more efficient way. Predictors for DRP appear to be easily assessable in clinical settings, providing a promising approach for the integration of the tool into clinical practice, once external validation is conducted. Currently, external validation of the tool across various settings in Austria is planned.

## Supporting information

**S1 Table. TRIPOD Checklist, Prediction Model Development and Validation.**
(DOCX)

**S1 Appendix. Full details of tested predictors for the model development.**
(DOCX)

**S2 Appendix. Boxplots and barplots of participant characteristics.**
(DOCX)

**S2 Table. Correlations and web code for the variable selection.**
(DOCX)

**S3 Table. Model summary.**
(DOCX)

**S4 Table. Cut-off values of the model.**
(DOCX)

## Acknowledgments

The authors thank the participating physicians of the pre-anaesthesia clinic, service users and pharmacists for their support and willingness to contribute to this project. We express our gratitude to Dr. Martin Wolkersdorfer and Dr. Ulla Porsche (Landesapotheke Salzburg, Müllner Hauptstraße 50, 5020 Salzburg) for including the pharmaceutical personnel.

The study has been presented by Stephanie Clemens on the 13th Pharmaceutical Care Network Europe (PCNE) Working Conference 8-11th February 2023 in Hillerod, Denmark. An abstract was published in the International Journal of Clinical Pharmacy

## Author contributions

**Conceptualization:** Stephanie Clemens, Olaf Rose, Georg Zimmermann, Peter Gerner, Maria Flamm.

**Data curation:** Stephanie Clemens, Clara Simon.

**Formal analysis:** Wanda Lauth, Georg Zimmermann.

**Investigation:** Wanda Lauth, Georg Zimmermann.

**Methodology:** Stephanie Clemens, Georg Zimmermann.

**Project administration:** Stephanie Clemens, Olaf Rose, Peter Gerner, Christina Dückelmann, Johanna Pachmayr.

**Supervision:** Peter Gerner, Maria Flamm, Johanna Pachmayr, Olaf Rose.

**Validation:** Wanda Lauth, Georg Zimmermann.

**Visualization:** Stephanie Clemens, Wanda Lauth.

**Writing – original draft:** Stephanie Clemens, Wanda Lauth.

**Writing – review & editing:** Stephanie Clemens, Clara Simon, Wanda Lauth, Olaf Rose, Georg Zimmermann, Peter Gerner, Christina Dückelmann, Maria Flamm, Johanna Pachmayr.

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
