## [Decision Letter · Decision Letter 0]

22 Dec 2024

PONE-D-24-34407

Development and validation of a risk prediction tool for drug-related problems in pre-operative elective surgical patients (mediPORT): a case-control study

PLOS ONE

Dear Dr. Clemens,

Thank you for submitting your manuscript to PLOS ONE. After careful consideration, we feel that it has merit but does not fully meet PLOS ONE’s publication criteria as it currently stands. Therefore, we invite you to submit a revised version of the manuscript that addresses the points raised during the review process.

We look forward to receiving your revised manuscript.

Kind regards,

Zhaoqing Du, Ph.D

Academic Editor

PLOS ONE

Journal Requirements:

2. For studies involving third-party data, we encourage authors to share any data specific to their analyses that they can legally distribute. PLOS recognizes, however, that authors may be using third-party data they do not have the rights to share. When third-party data cannot be publicly shared, authors must provide all information necessary for interested researchers to apply to gain access to the data. (https://journals.plos.org/plosone/s/data-availability#loc-acceptable-data-access-restrictions)

Reviewers' comments:

Reviewer's Responses to Questions

**Comments to the Author**

1. Is the manuscript technically sound, and do the data support the conclusions?

Reviewer #1: Partly

Reviewer #2: Yes

Reviewer #3: Yes

2. Has the statistical analysis been performed appropriately and rigorously? 

Reviewer #1: Yes

Reviewer #2: Yes

Reviewer #3: Yes

3. Have the authors made all data underlying the findings in their manuscript fully available?

Reviewer #1: Yes

Reviewer #2: No

Reviewer #3: Yes

4. Is the manuscript presented in an intelligible fashion and written in standard English?

Reviewer #1: Yes

Reviewer #2: Yes

Reviewer #3: Yes

5. Review Comments to the Author

Reviewer #1: Too long abstract. You can shorten it. ‎

Too many old ref. you can use the recent ref (within the last 5 years)‎.

‎Please use (google scholar and Refseek) search engines then set it since 2020‎

you can use my suggestions

Reviewer #2: Thank you so much for your exceptional work and dedication in addressing all the comments provided by the previous reviewers. Your thorough and thoughtful approach to incorporating the feedback has undoubtedly enhanced the overall quality and clarity of the work. It is clear that you have invested significant effort into ensuring that every aspect of the reviewers' suggestions has been considered and addressed comprehensively.

However, I suggest that to make a plan to validate the prediction model in more hospitals with different study design.

Reviewer #3: Most research aspects are performed perfectly.

The research idea is novel. The objectives were clarified and listed. The methodology chosen and applied in good way and was sufficient to explore and answer the research questions and achieving the research objectives. The findings showed clearly in the result section. The study findings and thoughts discussed optimally.

I suggest to expand the introduction section to cover the relevant literature adequately. In addition, proofreading for the full manuscript is required to address few grammatical mistakes existing within the submitted manuscript.

No further comments is needed. I endorse publication with minor amendments.

Best wishes.

6. PLOS authors have the option to publish the peer review history of their article (what does this mean?). If published, this will include your full peer review and any attached files.

Reviewer #1: **Yes: **Hazim Alhiti

Reviewer #2: **Yes: **Prof. Yaser Mohammed Al-Worafi, PhD

Reviewer #3: No

---

## [Author Response · Author response to Decision Letter 1]

14 Jan 2025

Dear Editor, dear Reviewers,

we sincerely appreciate your valuable insights and the time you dedicated to reviewing our manuscript. Your suggestions have significantly enhanced its quality. Following your recommendations, we have revised the manuscript accordingly and provided a detailed, point-by-point response. Our replies are in italics, and the revisions to the manuscript are highlighted in blue. For some of the suggestions, we would like to clarify the following:

• We have shortened the abstract

• We have updated references to include more recent sources and have gone from 35 to 39 references

• We have extent the introduction

• A language check was conducted by a natural speaker

We included the detailed answer of all comments in the document Response to Reviewers.

We hope that these changes to the manuscript will meet your expectations.

Sincerely,

Stephanie Clemens (Corresponding Author)

---

## [Decision Letter · Decision Letter 1]

26 Jun 2025

PONE-D-24-34407R1
Development and validation of a risk prediction tool for drug-related problems in pre-operative elective surgical patients (mediPORT): a case-control study
PLOS ONE

Dear Dr. Clemens,

Thank you for submitting your manuscript to PLOS ONE. After careful consideration, we feel that it has merit but does not fully meet PLOS ONE’s publication criteria as it currently stands. Therefore, we invite you to submit a revised version of the manuscript that addresses the points raised during the review process.

We look forward to receiving your revised manuscript.

Kind regards,

Jianhong Zhou

Staff Editor

PLOS ONE

Journal Requirements:

Reviewers' comments:

Reviewer's Responses to Questions

**Comments to the Author**

1. If the authors have adequately addressed your comments raised in a previous round of review and you feel that this manuscript is now acceptable for publication, you may indicate that here to bypass the “Comments to the Author” section, enter your conflict of interest statement in the “Confidential to Editor” section, and submit your "Accept" recommendation.

Reviewer #4: All comments have been addressed

Reviewer #5: (No Response)

2. Is the manuscript technically sound, and do the data support the conclusions?

Reviewer #4: Yes

Reviewer #5: Yes

3. Has the statistical analysis been performed appropriately and rigorously? 

Reviewer #4: Yes

Reviewer #5: Yes

4. Have the authors made all data underlying the findings in their manuscript fully available?

Reviewer #4: (No Response)

Reviewer #5: Yes

5. Is the manuscript presented in an intelligible fashion and written in standard English?

Reviewer #4: Yes

Reviewer #5: Yes

6. Review Comments to the Author

Reviewer #4: The study was great and would have significant contributions to the pharmacy practices.

The authors have responded to the previous reviewer’s comments. Most of the comments and research aspects were described clearly.

However, there is an additional minor comment from the file PONE-D-24-34407_R1_reviewer as follows:

1. Please describe in more detail the rationale for setting the sample case-control ratio of 1:4 (line 181 or 149 updated version

Reviewer #5: I would like to sincerely thank the authors for their thoughtful and rigorous work on this important topic. The manuscript reflects a high level of scholarly effort, methodological care, and clinical insight. The development of the mediPORT tool represents a meaningful contribution to improving medication safety in surgical care, and I commend the authors for their dedication to advancing patient-centered clinical pharmacy practice.

Strengths:

• The study addresses a significant gap in perioperative pharmacotherapy and patient safety.

• The use of a large, well-defined dataset and robust statistical modeling (including internal validation) enhances the credibility of the findings.

• The inclusion of both a comprehensive and a simplified model increases the tool’s applicability across diverse clinical settings.

Recommendations for Minor Revision:

• The observed higher DRP risk in CKD stage 3 compared to stage 5 warrants further discussion. The authors may consider elaborating on potential clinical explanations or stratifying by drug class in future analyses.

• Ensure all figures are high-resolution and clearly labeled. Figure 4 (mediPORT tool) could benefit from a more intuitive visual layout

7. PLOS authors have the option to publish the peer review history of their article (what does this mean?). If published, this will include your full peer review and any attached files.

Reviewer #4: No

Reviewer #5: **Yes: **Tsegaye Melaku

---

## [Author Response · Author response to Decision Letter 2]

14 Jul 2025

Dear Reviewers,

we sincerely appreciate your valuable insights and the time you dedicated to reviewing our revised manuscript. Your suggestions have significantly enhanced its quality. Following your recommendations, we have revised the manuscript accordingly and provided a detailed, point-by-point response. For the suggestions, we would like to clarify the following:

• We have revised Figure 4

• We have provided a more detailed explanation of the rationale behind choosing a 1:4 case-control ratio

• We have added a statement to the discussion highlighting the potential for future analyses involving drug class stratification

We hope that these changes to the manuscript will meet your expectations.

Sincerely,

Stephanie Clemens

---

## [Decision Letter · Decision Letter 2]

4 Aug 2025

Development and validation of a risk prediction tool for drug-related problems in pre-operative elective surgical patients (mediPORT): a case-control study

PONE-D-24-34407R2

Dear Dr. Stephanie Clemens,

We’re pleased to inform you that your manuscript has been judged scientifically suitable for publication and will be formally accepted for publication once it meets all outstanding technical requirements.

Kind regards,

Naeem Mubarak, PhD

Academic Editor

PLOS ONE

Additional Editor Comments (optional):

The manuscript needs no further revisions

Reviewers' comments:

Reviewer's Responses to Questions

**Comments to the Author**

1. If the authors have adequately addressed your comments raised in a previous round of review and you feel that this manuscript is now acceptable for publication, you may indicate that here to bypass the “Comments to the Author” section, enter your conflict of interest statement in the “Confidential to Editor” section, and submit your "Accept" recommendation.

Reviewer #4: All comments have been addressed

Reviewer #5: All comments have been addressed

2. Is the manuscript technically sound, and do the data support the conclusions?

Reviewer #4: Yes

Reviewer #5: Yes

3. Has the statistical analysis been performed appropriately and rigorously? 

Reviewer #4: Yes

Reviewer #5: Yes

4. Have the authors made all data underlying the findings in their manuscript fully available?

Reviewer #4: Yes

Reviewer #5: Yes

5. Is the manuscript presented in an intelligible fashion and written in standard English?

Reviewer #4: Yes

Reviewer #5: Yes

6. Review Comments to the Author

Reviewer #4: The Authors have revised the manuscript as suggested. The topics is important for the implementation of pharmacy practices.

The paper is recommended for publication.

Reviewer #5: Thank you, I am pleased to note that all of my feedback from the previous version has been thoroughly addressed. Thus, I recommend the acceptance of the article.

7. PLOS authors have the option to publish the peer review history of their article (what does this mean?). If published, this will include your full peer review and any attached files.

Reviewer #4: No

Reviewer #5: **Yes: **Tsegaye Melaku

---

## [Editor Report · Acceptance letter]

PONE-D-24-34407R2

PLOS ONE

Dear Dr. Clemens,

I'm pleased to inform you that your manuscript has been deemed suitable for publication in PLOS ONE. Congratulations! Your manuscript is now being handed over to our production team.

Kind regards,

on behalf of

Dr Naeem Mubarak

Academic Editor

PLOS ONE